# A Household-Based Survey of Iodine Nutrition in Moroccan Children Shows Iodine Sufficiency at the National Level But Risk of Deficient Intakes in Mountainous Areas

**DOI:** 10.3390/children8030240

**Published:** 2021-03-19

**Authors:** Laila El Ammari, Naima Saeid, Anouar Talouizte, Hasnae Gamih, Salwa Labzizi, Jamila El Mendili, Anass Rami, Mohamed Idrissi, Kaoutar Benjeddou, Fatima ezzahra Zahrou, Mohamed Elmzibri, Amal Tucker Brown, Ayoub Al-Jawaldeh, Abdelhakim Yahyane, Michael Bruce Zimmerman, Khalid El Kari, Hassan Aguenaou

**Affiliations:** 1Ministry of Health, Rabat 10090, Morocco; ca.anouar@gmail.com (A.T.); hasnae.gamih@gmail.com (H.G.); salwalabzizi@gmail.com (S.L.); elmendilij@yahoo.fr (J.E.M.); hakimdocy@yahoo.fr (A.Y.); 2Ibn Tofail University-CNESTEN, Joint Research Unit in Nutrition and Food, RDC-Nutrition AFRA/IAEA, Rabat-Kénitra 1400, Morocco; saeid_na@yahoo.com (N.S.); anassrami13@gmail.com (A.R.); idrissom@gmail.com (M.I.); kaoutar.benjed@hotmail.com (K.B.); fzahrou@yahoo.com (F.e.Z.); elmzibri@gmail.com (M.E.); khalidelkari73@gmail.com (K.E.K.); aguenaou.hassan@uit.ac.ma (H.A.); 3Iodine Global Network, Rabat 10090, Morocco; amal.tuckerbrown@gmail.com; 4World Health Organisation, Regional Office for Eastern Mediterranean, P.O. Box 7608, Nasr City 11371, Cairo, Egypt; aljawaldeha@who.int; 5ETH Zürich, Laboratory of Human Nutrition, Institute of Food Nutrition and Health, Department of Health Science and Technology, ETH Zurich, 8092 Zürich, Switzerland; michael.zimmermann@hest.ethz.ch

**Keywords:** iodine, deficiency, iodized salt, children, urinary iodine concentration, high-altitude, Morocco

## Abstract

Historically, mountainous areas of Morocco have been affected by endemic goiter and severe iodine deficiency. In 1995, Morocco legislated salt iodization to reduce iodine deficiency. There has been no national survey of iodine nutrition in school-age children for nearly 3 decades. Our aim was to assess iodine nutrition in a national sample of 6–12-year-old children in Morocco to inform the national salt iodization strategy. In this cross-sectional household-based survey, we randomly recruited healthy 6–12-year-old children from 180 clusters in four geographic zones (north and east, central, north and south) covering the 12 regions of Morocco. A questionnaire was completed, including socio-economic status and parental level of education. In addition, anthropometric measurements were taken to assess nutrition status, and a spot urine sample was collected to measure urinary iodine concentration (UIC). A total of 3118 households were surveyed, and 1043 eligible children were recruited, 56% from urban areas and 44% from rural areas. At the national level, the percentage of surveyed samples with UIC < 50 μg/L was 21.6% (19.2%; 24.2%), which exceeds the WHO suggestion of no more than 20% of samples below 50 μg/L, despite an adequate level of median urinary iodine concentration (mUIC) at 117.4 µg/L (110.2; 123.3). There were no statistically significant differences in mUIC comparing urban vs. rural areas and socio-economic status. However, the mUIC was significantly lower in the central (high-altitude non-coastal) zone (*p* < 0.004), where the mUIC (95% CI) was deficient at 89.2 µg/L (80.8; 102.9). There was also a significant difference in the mUIC by head of household education level (*p* = 0.008). The mUIC in Moroccan children >100 µg/L indicates iodine sufficiency at the national level. However, the percentage of surveyed samples with UIC < 50 μg/L above suggests that a significant proportion of children remain at risk for iodine deficiency, and it appears those at greatest risk are residing in the central (high altitude non-coastal) zone. A national level mUIC value may conceal discrepancies in iodine intake among different sub-groups, including those defined by geographic region.

## 1. Introduction

Iodine deficiency (ID) has multiple adverse effects on growth and development in humans, collectively termed iodine deficiency disorders (IDDs) [1]. Even mild to moderate ID in school-aged children (SAC) can impair cognitive function [2]. Because ID continues to affect large populations, particularly in Africa and South Asia, it is an important preventable cause of intelligence quotient (IQ) point loss and impaired school learning [3,4]. Salt iodization is an effective and sustainable strategy for the prevention of IDD [5]. Salt iodization is recommended because salt is used in nearly all diets and its intake is fairly consistent throughout the year. Iodization of salt is simple and inexpensive, and its addition does not affect salt color or flavor [5,6].

Morocco, a country in North Africa with a population of ca. 36 million, has legislated salt iodization as the national strategy against IDD [7]. In 1995, mandatory iodization of table salt at 80 ± 10 mg of iodine/kg salt was introduced and subsequently adjusted in 2009 to 15–40 mg/kg [8]. Several previous studies in Morocco have assessed the iodine nutrition of children. In 1993, a national survey among SAC reported a median urinary iodine concentration (mUIC) of 86 μg/L and a mean goiter prevalence of 22% (95% CI: 20.0–24.1), but prevalence was highly variable, ranging from 0% to 77.4% in different regions [7]. Subsequently, smaller regional studies in the mountains of Morocco reported that the prevalence of goiter among SAC was as high as 67% [9] and mUIC was <20 μg/L, indicating severe IDD [10]. A study of SAC in another rural mountainous area reported that the mUIC of SAC was 59.6 µg/L, suggesting mild ID [11].

Because >90% of dietary iodine eventually appears in urine [12], UIC is an excellent indicator of recent iodine intake. For national surveys of iodine nutrition, WHO recommends that UIC should be measured in spot urine specimens from a representative sample of SAC and expressed as the median, in μg/L [5], and this can be used to classify a population’s iodine status [5]. It has been nearly 3 decades since the last national survey of iodine nutrition in Moroccan SAC [7], and, considering previous reports of severe IDD among SAC [9,10], updated data are urgently needed. Therefore, the aim of our study was to assess the current iodine nutrition in a nationally representative sample of 6–12-year-old SAC in Morocco by measuring the UIC in spot urine samples to inform the national salt iodization strategy.

## 2. Materials and Methods

### 2.1. Study Design and Participants

This household-based survey was taken from March to June 2019. It was conducted at the national level in four geographic zones (North and East zone 1: coastal, mountainous; West zone 2: coastal, non-mountainous; Central zone 3: high altitude, non-coastal; South zone 4: coastal, mountainous, desert), covering the 12 regions of Morocco. A total of 180 clusters were selected using the probability proportionate to size sampling approach, as recommended by the WHO [13]. Within each cluster, households were randomly selected based on the enumeration sheet completed the day before the survey. A sample of 20 households was selected according to the systematic one-point starting method with equal probability. In total, 3118 households were surveyed (60.4% in urban areas and 39.6% in rural areas). Our sample size was based on WHO recommendations for a sample size of at least 1200 in national surveys of iodine nutrition in SAC [5]. Based on data from the Moroccan STEPS-2018 survey [14], we anticipated that we would be able to recruit about 1600 children using our household-based recruitment strategy, and, accounting for attrition and incomplete sample collection/analysis, this would provide at least 1200 SAC.

Within each selected household, a child aged 6–12 years was recruited for the survey if he/she was present in the household. If several children met the inclusion and exclusion criteria at one household, a random draw based on Kish’s table was carried out by the team supervisor. In a meeting with the family, the study’s purpose was explained and informed consent was obtained from the child’s parents. Exclusion criteria included ages younger than 6 years or older than 12 years and having a chronic or severe illness requiring hospitalization or treatment. The study received ethical approval from the Faculty of Medicine and Pharmacy, Mohammed University in Rabat—Morocco (Ethical Approval number 321; 3 April 2017).

### 2.2. Study Procedures

A questionnaire was filled out within each selected household, including the age of child, gender, socio-economic status, and parental level of education. Anthropometric measurements were taken according to standard procedures [15]. Bodyweight was measured to the nearest 0.1 kg using an electronic scale (Seca GmbH, Hamburg, Germany). Height was measured to the nearest 0.1 cm using a portable Shorr Board (formerly Shorr Productions, LLC; Olney, MD, USA) without shoes. Waist circumference (WC) was measured at a 0.1 cm threshold using an inelastic tape measure. A mid-stream spot urine sample was collected into clean, tightly capped containers, stored at −20 degrees until analysis.

### 2.3. Laboratory and Data Analyses

Body mass index (BMI) was calculated as a ratio of weight in kg divided by height in meters squared. According to the World Health Organization, stunting and thinness were defined as Height-for-Age z-score (HAZ) and Body Mass Index-for-Age z-score (BAZ); Z-scores <−2, respectively [16]. Overweight was defined as Body Mass Index-for-Age (BAZ); Z-scores > +2, and obesity as Body Mass Index-for-Age (BAZ); Z-scores > +3. Z-scores were calculated using software Anthro+ [16].

Urinary iodine was determined spectrophotometrically using the Sandell–Kolthoff reaction. Iodine status was classified as follows: excess iodine intake, mUIC >300 µg/L; adequate iodine intake, 100–299 µg/L; mild iodine deficiency, 50–99µg/L; moderate iodine deficiency, 20–49µg/L, and severe iodine deficiency, <20µg/L [17,18,19]. Urinary iodine values were also expressed according to the urinary creatinine concentration to reflect 24-hr urinary iodine excretion. Statistical analyses were performed using the Statistical Package for the Social Sciences (SPSS, version 20.0). Variables that were normally distributed were presented as the mean (SD). Variables not normally distributed were presented as the median (interquartile range (IQR)). Nominal variables were presented as frequencies (95% confidence interval (CI)). Comparisons of frequencies were done using Chi-square, comparisons of normally distributed data were done using Student’s t-test, and comparisons of nonnormally distributed data were done using Kruskal–Wallis tests. *p*-values < 0.05 was considered statistically significant.

## 3. Results

We recruited 1043 eligible children from the sampled households. Geographic and socio-demographic characteristics of the studied population are shown in Table 1. In total, 56.3% were from urban areas and 43.7% were from rural areas; 50.5% were boys and 49.5% were girls. In total, 40.1% of children were from households belonging to the lowest socio-economic tercile and 27.8% were from the highest tercile. Age and anthropometric characteristics are presented in Table 2. Mean weight, height, and BMI were significantly higher in children from urban areas compared to rural areas (*p* < 0.05). The overall prevalence of overweight children and children with obesity was 13.5% and 5%, respectively. Overweight children and obesity were more common in urban areas compared to rural areas (14.2% vs. 12.5%) and (7.3% vs. 1.9%), respectively (*p* < 0.001 and *p* < 0.001). However, the prevalence of stunting (HAZ < −2SD) was significantly higher in rural areas compared to urban areas (15% vs. 8.1%) (*p* = 0.002).

The mUIC at the national level, according to the area of residence (urban vs. rural), and by gender is shown in Figure 1. The percentage of surveyed samples with UIC < 50 μg/L was 21.6% (19.2%; 24.2%), which exceeds the WHO suggestion of no more than 20% of samples below 50 μg/L, despite an adequate level of mUIC (95% CI) in the total sample, which was 117.4 µg/L (110.2; 123.3).

There were no statistically significant differences in mUIC by gender: the mUIC (95% CI) in boys and girls was 116.1 µg/L (107.3; 125.6) and 118.8 µg/L (102.1; 128.5), respectively. There were no statistically significant differences in mUIC by urban vs. rural areas: the mUIC (95% CI) in urban and rural clusters was 119.0 µg/L (109.9; 128.8) and 115.3 µg/L (102.1; 123.3), respectively.

Table 3 shows the mUIC and the number and prevalence of UIC < 50 µg/L, according to head-of-household education level, geographic zone, and socio-economic status. There were no significant differences in mUIC according to socio-economic status. However, the mUIC was significantly lower in zone 3 (the high attitude non-coastal zone) compared to the other zones (*p* < 0.004). In the central (high attitude non-coastal) zone, children were mildly iodine deficient, with a mUIC (95% CI) of 89.2 (80.8; 102.9) (Table 3). Additionally, the mUIC was significantly lower among households with a preparatory household’s head education with a mUIC (95% CI) of 93 (79.7; 112.3).

## 4. Discussion

The main finding of this study is that the percentage of surveyed samples with UIC <50 μg/L was 21.6% (19.2%; 24.2%), which exceeds the WHO (2007) [4] and UNICEF (2018) [20] recommendation of no more than 20% of samples at <50μg/L in UIC surveys of children, despite an adequate level of mUIC (95% CI) in the total sample, which was 117.4 µg/L (110.2; 123.3). These findings suggest that, despite iodine sufficiency at the national level, a small but significant proportion of children in Morocco may remain at risk for IDD.

A strength of this study was that we assessed the adequacy of iodine intake among different subsets of the population of SAC that may be more vulnerable to deficiency. A national level mUIC value may conceal discrepancies in iodine intake among different sub-groups, including geographic region and/or socio-economic status [20]. Our stratified analyses suggest that iodine intake is comparable across rural and urban areas and across households with varying socioeconomic status but not parental education. This equitable distribution of iodine intake suggests that the rural poor in Morocco are not at higher risk of ID, an important finding. However, our geographic zone analyses suggest that children residing in the central (high altitude non-coastal) zone are at higher risk of ID than those residing at lower non-coastal altitudes. This finding is in agreement with previous studies in other countries reporting an increase in the prevalence of ID with increasing altitude [21,22,23,24]. Historically, before the introduction of iodized salt, the mountain valleys of the Alps, Himalayas, and Andes were areas with severe endemic goiter and IDD due to glaciation and leaching of iodine from soils [12]. Our data suggest that reaching children in high altitude non-coastal areas in Morocco with adequate iodine remains a challenge, such as communities in the Atlas Mountains in the south [9] and/or the Rif mountains in the north [10]. Based on our data, a future focus of the Moroccan iodized salt program should improve coverage of adequately iodized salt, particularly in these high-altitude non-coastal areas that do not have access to intrinsic sources of iodine, such as salt-water fish, seafood, and milk and dairy products from animals who consume iodine-rich feeds and fodder.

The salt industry sector in Morocco is poorly structured and organized, and many salt producers are not iodizing their salt; indeed, an assessment of the iodine content of salt found that only 7.5% of salt was adequately iodized (15–40 ppm) and 25.7% of salt was not iodized (<5 ppm) (results under review). The situation is further complicated by insufficient official control, an informal sector without any traceability of the origin of the salt, and the coexistence in the market of iodized salt and non-iodized salt. Thus, the Moroccan national iodine program should improve and strengthen regulatory monitoring and ensure the salt industry’s capacity to consistently produce adequately iodized salt.

Given the growing proportion of salt coming from processed foods in Morocco and the national policy to reduce salt consumption by 10% by 2025 [25], it may be important to extend mandatory salt iodization to include all salt used in the food industry. This could allow the salt iodization program to continue to control iodine deficiency, while the envisioned reduction in overall salt intake could contribute to the prevention of non-communicable diseases [26]. In transitioning and industrialized countries, because more than 80% of salt consumption is from purchased processed foods, if only household salt is iodized, it may not supply adequate iodine [4]. Thus, to successfully control iodine deficiency, it is critical to convince the food industry to use iodized salt in their products. Fortunately, iodine at ppm levels in foods does not cause any sensory changes, and, in most countries, the price difference between iodized and non-iodized salt is negligible, so there are no major barriers to its use in processed foods [12]. In Denmark and the Netherlands, nearly all salt used by the baking industry is iodized, which controls iodine deficiency. Switzerland’s long-running iodized salt program has been successful because 60% of salt used by the food industry is iodized on a voluntary basis [27]. The current global push to reduce salt consumption to prevent chronic diseases and the policy of salt iodization to control ID do not conflict: iodization methods can fortify salt to provide adequate iodine even if per capita salt intakes are reduced to <5 g/day, as long as all salt consumed is iodized [26,28].

In addition, iodine supplementation of animal feed could also be considered because milk and eggs can be good dietary sources of iodine. In areas with poor iodine soils (especially high-altitude areas), groundwater and plants are typically deficient in iodine [12]. Animals that graze on these soils are at risk for iodine deficiency, and the resulting animal food products, such as milk, meat, and eggs, may be particularly low in iodine. In other countries [29,30], iodine supplementation of animal feed has also improved animal health and increased the production performance of animals. It can contribute to the control of iodine deficiency in human populations [31]. However, such an animal supplementation strategy is meant to complement salt iodization, not replace it.

Although this study was not designed to provide a national estimate of stunting or overweight children/obesity in Moroccan children, our analysis of the anthropometric measurements in this national sample provides important insights. Our data suggest a double burden of malnutrition among the studied children. Indeed, 11.5% of children were stunted, but at the same time, 13.5% of the children were overweight and 5% were obese. The comparison between urban and rural areas showed no significant differences in wasting or overweight children. However, the prevalence of stunting was significantly greater in rural areas, and the prevalence of obesity was greater in urban areas. Our findings are consistent with a 2015 study in children attending public schools in Azilal (central Morocco), where the prevalences of stunting and underweight children were 8.5% and 3.4%, respectively [32].

## 5. Conclusions

The overall percentage of surveyed children in Morocco with UIC < 50 μg/L is greater than 20%, which suggests that despite an adequate mUIC, a proportion of children in Morocco remain at risk for iodine deficiency, and it appears the populations at greatest risk are those residing in high non-coastal altitude zones. Future efforts should focus on reaching these areas. In addition, ensuring that all salt for animal and human consumption is iodized should be considered and would likely improve the iodine status of the population. Additional research is needed to determine the dietary sources of iodine in iodine-sufficient areas of Morocco. Encouraging the food industry to use iodized salt in processed foods and strengthening the regulatory monitoring of iodization at the production sites to ensure adequate iodization of all table salt may also be valuable to ensure adequate iodine for all Moroccan children.

## Figures and Tables

**Figure 1 children-08-00240-f001:**
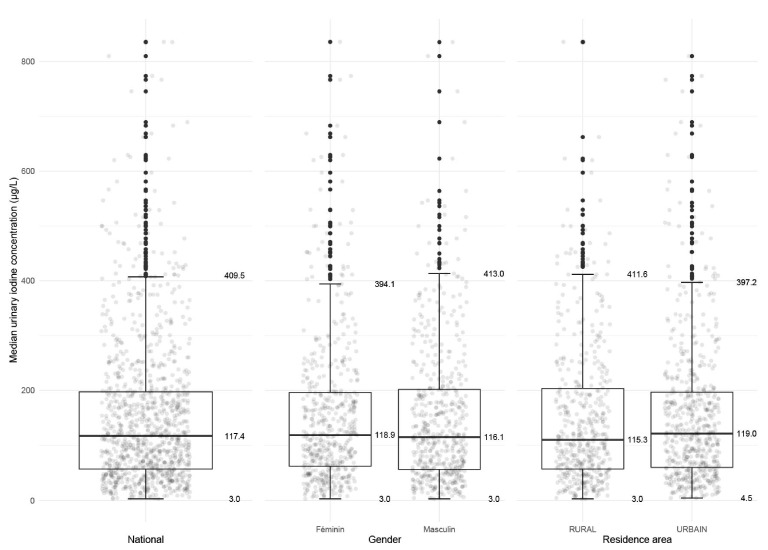
The median urinary iodine in all subjects by gender and residence area.

**Table 1 children-08-00240-t001:** Socio-demographic characteristics of the studied population.

	Sample Size (*n* = 1043)	% Weighting	95% CI
**Residence Area**			
Urban	570	56.3%	53.2%; 59.2%
Rural	473	43.7%	40.8%; 46.8%
**Gender**			
Boys	527	50.5%	54.2%; 51.1%
Girls	516	49.5%	51.9%; 48.9%
**Geographies Areas**			
North and East zone 1 coastal, mountainous	269	26.9%	24.3%; 29.7%
West zone 2 coastal, non-mountainous	384	40.6%	37.6%; 43.6%
Central zone 3 High altitude, non-coastal	277	22.7%	20.2%; 25.3%
South zone 4 coastal, mountainous, desert	113	9.7%	8.0%; 11.6%
**Economic Status**			
Tercile 1	432	40.1%	37.1%; 43.1%
Tercile 2	329	32.1%	29.3%; 35.0%
Tercile 3	282	27.8%	25.1%; 30.6%

Data are presented as proportion (95% confidence interval) based on 1000 bootstrap samples.

**Table 2 children-08-00240-t002:** Age and anthropometric characteristics of the studied population.

	National	Gender		*p* Value	Residence Area		*p* Value
		Boy	Girl		Urban	Rural	
Age, month Mean ± SD	107	108 ± 23.4	106 ± 22.3	0.251 ^a^	107 ± 22.8	106 ± 22.9	0.525 ^a^
Weight, Kg Mean ± SD	28.2	28.1 ± 7.62	28.3 ± 9.06	0.704 ^a^	29.3 ± 9.2	26.7± 6.8	<0.001 ^a^
Height, cm Mean ± SD	128.9	129.3 ± 11.7	128.4 ± 12.3	0.254 ^a^	130.4 ± 12.5	126.9 ± 11	<0.001 ^a^
BMI for age, Z-score Mean ± SD	0.0	0.0 ± 1.19	0.08 ± 1.09	0.232 ^a^	0.1 ± 1.2	−0.0801	0.003 ^a^
HAZ Z-score Mean ± SD	0.00	−0.5 ± 1.3	−0.56 ± 1.3	0.603 ^a^	−0.3 ± 0.9	1.3 ± 1.3	<0.001 ^a^
HAZ <−2 SD %	11.1	10.8	11.0	0.104 ^b^	8.1	15.0%	0.002 ^b^
BAZ < −2%	2.8	3.1	2.4	0.935 ^b^	2.2	3.5	<0.001 ^b^
BAZ> + 1ET et ≤ + 2ET %	13.5	12.8	14.2		14.2	12.5	
BAZ > + 2ET%	5.0	4.9	5.1		7.3	1.9	

^a^ Test de Student, ^b^ Test Chi-square; body mass index (BMI); height for age z-score (HAZ); BMI for age z-score (BAZ). Results are presented as mean ± SD and frequency (proportion).

**Table 3 children-08-00240-t003:** Median urinary iodine concentration (mUIC) and the number and percentage of surveyed samples with UIC < 50 μg/L according to head-of-household education level, geographic zone, and socio-economic status.

Geographic Zone/Demographic Characteristics		Median of UIC µg/L				UIC <50 µg		
		Median	N	95% CI of the Median	*p*-Value ^a^	%	95% CI of the %	*p*-Value ^b^
Zones	North and East zone 1 coastal, mountainous	135.1	269	122.2; 154.1	<0.001	14.5	10.9; 19.1	<0.004
	West zone 2 coastal, non-mountainous	118.6	384	112.1; 134.3		22.5	18.7; 26.7	
	Central Zone 3 high altitude, non-coastal	89.2	277	80.8; 102.9		27.1	21.8; 33.0	
	South Zone 4 coastal, mountainous, Desert	121.9	113	90.6; 155.7		24.7	17.1; 33.8	
Level of household’s head education	Primary	126.6	298	118.6; 140.4	0.0088	18.1	14.1; 23.3	0.02
	Preparatory	93	105	79.7; 112.3		26.3	18.6; 34.4	
	Secondary	155.8	128	127.9; 176.5		13.5	8; 20.4	
	Superior	112	54	94.5; 162.2		18.9	10.8; 28.8	
	Illiterate	105	457	94.6; 119.7		24.7	21; 28.7	
Economic status	Tercile 1	113.2	432	100.5; 129.0	0.069	22.2	18.3; 26.2	0.219
	Tercile 2	112.1	329	96.0; 122.6		23.9	19.6; 28.7	
	Tercile 3	128.4	282	115.9; 140.9		18.2	14.2; 23.1	

^a^ Kruskal-Wallis Test, ^b^ Chi-square Test, CI: Confidence Interval.

## Data Availability

The data presented in this study are available on request from the corresponding author.

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
