# Peer review of "A Household-Based Survey of Iodine Nutrition in Moroccan Children Shows Iodine Sufficiency at the National Level But Risk of Deficient Intakes in Mountainous Areas"

_children, 2021, doi:10.3390/children8030240_

Round 1

Reviewer 1 Report

  1. A main finding of the current study is the unequal iodine nutrition in different geographic zones, with the median urinary iodine concentration (UIC) significantly lower in the “high altitude zone” compared to other zones. However, the characteristics of the “high altitude zone”, such as its geographic appearance or the use of iodized salt, are not described.

 Actually, there is valuable information to be found in the supplemental material, such as the map delineating the four zones, the use of iodized salt as shown in Supplemental Table 2, and the source of information on iodized salt in Supplemental Table 3. These materials could be added to the main text.

  1. In describing the iodine nutrition status of a population, the median UIC presents the central tendency, while the percentage of each UIC categories such as <50 μg/L, 50-100 μg/L, etc. relates to the distribution. Since one single UIC data cannot be used to define each individual’s iodine status, statements like - “prevalence of UIC <50 μg/L was xx%’ (in Abstract,line 33; Results, line 140, line 149-150; Discussion, line 170; Conclusion, line 240) are inappropriate. The intended meaning may be better expressed as “the percentage of surveyed samples with UIC < 50 μg/L was xx% , which exceeds the WHO suggestion of no more than 20% of samples below 50 μg/L, despite of an adequate level of median UIC.”

  1. In Table 3, the p-value for the comparison of median UIC categorized by household head’s level of education is 0.008. Why is this described as having “no significant difference” (Results, line 160)?

Author Response

Dear reviewer

We are grateful for your constructive comments of our manuscript 1136589. We have systematically responded to all your comments, and we have rewritten some sections of the manuscript to fully address your remarks. The language has also been corrected to improve readability and clarity of content. We have addressed your comments point by point.

Reviewer 2 Report

A HOUSEHOLD-BASED SURVEY OF IODINE NUTRITION IN MOROCCAN CHILDREN SHOWS IODINE SUFFICIENCY AT THE NATIONAL LEVEL BUT RISK FOR DEFICIENT INTAKES IN MOUNTAINOUS AREAS

The authors have done excellent job doing one of the laborious study. I have some few suggestions and edit that could improve this paper:

  1. Abstract, lines 28-30: The statement is winding – too long and so makes it unclear. There are three statements combined – questionnaire, anthropometric, and urine collection. Suggesting to split them.
  2. Abstract, line 31 starts with a number. I think it is standard practice not to start a statement with a number. I will spell out the number or add “a total of” to start that phrase.
  3. Abstract, line 34: spell out mUIC on first use in the abstract.
  4. Methods, section 2.3, Laboratory and data analysis, line 113: Put reference number 15 at the end of line 113 too.
  5. Methods, section 2.3, Laboratory and data analysis, line 111: Edit “body weight in kg by squared”. It is missing the word “divided”. Suggesting to edit it to read “body weight in kg divided by height in meters squared”.
  6. Table 2: Font size of the alphabetic superscript labels are too small. I think increasing the font size will improve visibility.
  7. Page 7, line 183: Something is definitely missing in “zones are be at higher”.
  8. Page 7-8, Lines 230-232: The information here suggest a “childhood double-burden of undernutrition-overnutrition in this population”. If you add this statement here or a similar statement, readers will not think the statement is conflicting, and would also help to emphasize the problem for policy making.
  9. It will enrich this paper a bit if the authors add a statement about some food sources of iodine available to this population, other than iodized salt, if any, and food sources that may contain goitrogens, etc., that can affect iodine nutrition and hence contribute to iodine deficiency, if such information is available in this population.
  10. It will enrich this paper a bit if the authors add statements to indicate that sea foods are among the best sources of iodine. However, Morocco is landlocked so salt iodization is one of the best means to eliminate IDDs.

Author Response

Dear Reviewer 

We are grateful for your constructive comments of our manuscript 1136589. We have systematically responded to all your comments, and we have rewritten some sections of the manuscript to fully address your remarks. The language has also been corrected to improve readability and clarity of content. We have addressed your comments point by point.
